# Bortezomib Eliminates Persistent *Chlamydia trachomatis* Infection through Rapid and Specific Host Cell Apoptosis

**DOI:** 10.3390/ijms23137434

**Published:** 2022-07-04

**Authors:** Ryota Itoh, Yusuke Kurihara, Michinobu Yoshimura, Kenji Hiromatsu

**Affiliations:** Department of Microbiology & Immunology, Faculty of Medicine, Fukuoka University, Fukuoka 814-0180, Japan; ykurihara@fukuoka-u.ac.jp (Y.K.); myoshimura@fukuoka-u.ac.jp (M.Y.); khiromatsu@fukuoka-u.ac.jp (K.H.)

**Keywords:** *Chlamydia* infection, anticancer drug, bortezomib, persistent infection

## Abstract

*Chlamydia trachomatis*, a parasitic intracellular bacterium, is a major human pathogen that causes millions of trachoma, sexually transmitted infections, and pneumonia cases worldwide. Previously, peptidomimetic inhibitors consisting of a hydrophobic dipeptide derivative exhibited significant inhibitory effects against chlamydial growth. Based on this finding, this study showed that both bortezomib (BTZ) and ixazomib (IXA), anticancer drugs characterized by proteasome inhibitors, have intensive inhibitory activity against *Chlamydia*. Both BTZ and IXA consisted of hydrophobic dipeptide derivatives and strongly restricted the growth of *Chlamydia* (BTZ, IC_50_ = 24 nM). In contrast, no growth inhibitory effect was observed for other nonintracellular parasitic bacteria, such as *Escherichia coli*. BTZ and IXA appeared to inhibit chlamydial growth bacteriostatically via electron microscopy. Surprisingly, *Chlamydia*-infected cells that induced a persistent infection state were selectively eliminated by BTZ treatment, whereas uninfected cells survived. These results strongly suggested the potential of boron compounds based on hydrophobic dipeptides for treating chlamydial infections, including persistent infections, which may be useful for future therapeutic use in chlamydial infectious diseases.

## 1. Introduction

*Chlamydiae* are major intracellular pathogens with a biphasic developmental cycle: a small, infectious but metabolically inactive elementary body (EB) and a metabolically active reticulate body (RB) [1]. After infecting a host cell, chlamydial RBs can replicate only in a parasitophorous vacuole termed inclusion inside the host cells. At the late phase of the infection cycle, RBs stop replication, redifferentiate to infective EBs, and eventually evade the host cell to infect the next target cells. *Chlamydiae* cause inflammation at the infection site, leading to sexually transmitted diseases, conjunctivitis, community-acquired pneumonia, arteriosclerosis, and proctitis [2].

Generally, antibiotics, such as macrolides or doxycycline, are used to treat against chlamydial infections, including genital tract infections, trachoma, pneumonia, and psittacosis, with a short course, and treatment has been successful in most cases [3,4]. However, not a few cases of treatment failure have been reported [5,6]. This is because *Chlamydia* becomes dormant in the inappropriate treatment of antibiotics or interferon-γ-mediated tryptophan starvation, suppressing their metabolism and making them insensitive to antimicrobial agents (persistent infection) [7,8,9]. The persistent infection of *Chlamydia* often lasts for years and is a major cause of infertility due to ascending oviductal edema [10]. In addition, the emergence of *Chlamydia* with drug resistance to macrolides has been reported [11], and the worldwide spread of such drug-resistant *Chlamydia* has been expected to threaten the future. For this reason, antibiotics with novel molecular mechanisms completely different from the existing antibiotics are always in demand.

Recently, a series of small-molecule compounds, such as z-Val-Phe-CHO (MDL-28170), z-Leu-Nle-CHO (calpeptin), or z-Phe-Ala-FMK, that consisting of an N-benzyloxycarbonyl group, hydrophobic dipeptides, and C-terminal aldehyde/fluoromethyl ketones, strongly inhibit the growth of *Chlamydia* [12]. These compounds are classified as protease inhibitors and are very common and reasonably inexpensive. However, there are some remaining issues. Relatively high concentrations are required for the antichlamydial activity (IC_50_ = ~15 μM). The chlamydial target molecule and the mechanism of growth inhibition against *Chlamydia* are still unknown.

This study showed that anticancer drugs bortezomib (BTZ) and ixazomib (IXA) have very strong antichlamydial activity. Furthermore, BTZ selectively kills *Chlamydia*-infected cells in an induced persistent infection state.

## 2. Results

### 2.1. Both BTZ and IXA but Not Delanzomib Strictly Suppress the Chlamydial Growth

Previous studies [12] searched for a series of commercially available compounds that fulfill the following points: a membrane-permeable cap structure, a hydrophobic dipeptide skeleton, and a strong reactive group at the C-terminus of the dipeptide. As a result, two antichlamydial agent candidates—namely, N-(2-pyrazinecarbonyl)-D-phenylalanine-L-leucine boronic anhydride (BTZ) and (R)-1-(2-(2,5-dichlorobenzamido)acetamido)-3-methylbutylboronic acid (MLN9708, IXA)—have been identified. BTZ and IXA are well-known proteasome inhibitors and approved anticancer drugs for treating multiple myeloma and mantle cell lymphoma [13,14,15]. First, whether both compounds affect chlamydial growth was examined. *Chlamydia*-infected HeLa cells were treated with 1 μM BTZ or IXA for 24 h. The control dimethyl sulfoxide (DMSO)-treated chlamydial inclusions showed well-grown phenotypes (Figure 1c), whereas BTZ- or IXA-treated chlamydial inclusions were apparently dwarfed (Figure 1d,e). Several particles, probably normal RBs, were in compound-treated small inclusions (Figure 1d,e, right). This study further investigated another preclinical anticancer drug, i.e., delanzomib [16,17], which has peptidyl boronate, as well as BTZ, but a nonhydrophobic dipeptide backbone (Figure 1f). Despite acting as the proteasome inhibitor, as well as BTZ or IXA, delanzomib did not show any effect against chlamydial inclusion (Figure 1g). The production of infectious progeny at 48 h postinfection (hpi) was investigated by counting the inclusion formation. BTZ and IXA demonstrated more than 100 times less EB production than the DMSO control (BTZ, 0.35 ± 0.25%; IXA, 0.82 ± 0.72%), even at a 1 μM low concentration (Figure 1h). In contrast, delanzomib showed very mild suppression at 1 μM (32.53 ± 6.26%). Only BTZ exhibited a mild cytotoxicity to the host cells; nevertheless, the chlamydial growth was dramatically reduced, indicating that BTZ directly inhibited the chlamydial growth rather than impaired the host cells (Figure 1i). BTZ also inhibited the *C. trachomatis* serovar D in HeLa cells (Appendix A). In addition, we confirmed that BTZ was effective against chlamydial growth in other host cell lines, such as HEp-2 or A549 (Appendix A).

### 2.2. Proteasome Inhibitors Other than BTZ or IXA Do Not Inhibit Chlamydial Growth

BTZ is an essentially intense proteasome inhibitor and approved for the single or combined treatment of multiple myeloma patients [18,19]. To examine the possibility that BTZ affects chlamydial growth via the proteasome inhibition of host cells, this study examined the antichlamydial effects of several well-known proteasome inhibitors that do not have a dipeptide backbone, such as MG132, MG115, proteasome inhibitor I (PSI), and carfilzomib (Figure 2). Immunostaining and the IFU assay revealed that MG132 (135.15 ± 63.17%), MG115 (193.07 ± 72.04%), and PSI (233.17 ± 85.35%) did not show any antichlamydial effects or, rather, promoted the production of infectious progeny (Figure 2b). Only carfilzomib apparently showed mild inhibition (54.22 ± 16.57%); however, the carfilzomib cytotoxicity was significantly higher than the other proteasome inhibitors (Figure 2c). 

### 2.3. BTZ Has Antichlamydial Activity Comparable to Existing Antimicrobial Drugs

A multipoint growth inhibition assay with *C. trachomatis* L2 was performed. As previously reported, z-hydrophobic dipeptide inhibitors MDL-28170 and z-FA-FMK strictly inhibited chlamydial growth at ≥25 μM [12]. However, it was conversely promoted at low concentrations (Figure 3a). The zero-equivalent point of each MDL-28170 and z-FA-FMK was 12 μM and 8.5 μM, with an IC_50_ of 16.6 μM and 14 μM, respectively (Figure 3a). Surprisingly, BTZ and IXA showed intense inhibition against *C. trachomatis*, with an IC_50_ of 24 nM and 70 nM, respectively (Figure 3a). Interestingly, BTZ did not show any growth inhibitory effect against other bacilli or cocci, such as *Escherichia coli*, *Staphylococcus aureus*, or *Klebsiella pneumoniae*, even at 50 μM (Figure 3b–d). To determine whether BTZ acts as a bactericidal or bacteriostatic against *Chlamydia*, *C. trachomatis*-infected HeLa cells were treated with BTZ at some periods. Cells treated with BTZ for a full 42 h (BTZ6-48) revealed a great inhibition of the chlamydial inclusion size compared to no treatment (Figure 3e–g, No treat). Cells treated with BTZ for only the former 18 h (BTZ6-24) showed moderate sizes of chlamydial inclusions (Figure 3e,h), suggesting the bacteriostatic effect of BTZ.

### 2.4. BTZ Arrests Chlamydial Replication but Does Not Induce the Persistent Form

To understand the mechanism of growth inhibition by BTZ, we analyzed the structure of chlamydial inclusion bodies using electron microscopy (EM). EM analyses of the infected control cells at 48 hpi revealed that chlamydial inclusions contained many chlamydial EBs, indicating late-stage RB–EB retransition (Figure 4, Control). In contrast, penicillin G (PCG)-treated infected cells showed several bizarre-looking inclusion bodies, which were compacted with aberrant RBs, a typical characteristic of PCG-induced persistent infection (Figure 4, PCG). Infected cells treated with BTZ for 48 h had a very small inclusion and contained very few chlamydial RBs (Figure 4, BTZ). IXA-treated chlamydial inclusion showed a very similar result compared to BTZ-treated inclusion (Figure 4, IXA). No aberrant RBs were observed in BTZ- or IXA-treated infected cells. 

### 2.5. BTZ Specifically Killed Persistently Chlamydia-Infected Host Cells

As described in Section 1, the persistent chlamydial infection is one of the most important problems of sexually transmitted infections. Despite this, the existing antimicrobial agents are ineffective against persistent infections that limit the metabolic activity. Therefore, whether BTZ treatment affects the persistent infection of *C. trachomatis* L2 in host cells was examined. *C. trachomatis*-infected host cells were treated with PCG to induce a persistent infection (from 3 to 48 hpi). Some of the infected cells were treated with PCG and BTZ for the last 24 h (Figure 5a). By the 45 h PCG treatment and 24 h after drug removal (recovery 24 h), weird shape inclusions and aberrant RBs were observed in the host cells, indicating persistent chlamydial infection (Figure 5b, left). Surprisingly, PCG- and BTZ-treated *Chlamydia*-infected cells were mostly shrunk and floated, apparently falling into cell death at 72 hpi (Figure 5b, right). At 96 h after drug removal (recovery 96 h), some PCG-treated chlamydial inclusions grew back normally (Figure 5c, left), whereas the PCG- and BTZ-treated *Chlamydia*-infected cells revealed further host cell death (Figure 5c, right). PCG- and BTZ-treated *C. trachomatis*-infected cells had positive dead cell indicator Zombie NIR^TM^ dye, although they were still adherent (Figure 5d, bottom), whereas noninfected cell and PCG-treated infected cells were Zombie NIR^TM^ dye-negative (Figure 5d, top). 

This study further confirmed that the number of live chlamydial inclusion bodies, including persistent chlamydial inclusions, decreased after BTZ treatment in a dose-dependent manner (Figure 5e). At 48 hpi, infectious EB production reaches the 10^8^ IFU score in the absence of chemical treatment but decreased to ~0.02% after PCG treatment (0.017 ± 0.004%) and was undetectable in PCG and BTZ treatment (Figure 5e, left; detection limit = 1 × 10^3^). Approximately 96 h after removal of the chemicals, the IFU score of PCG-treated *Chlamydia* was raised to nearly 100 times more than before the removal (1.74 ± 0.47% compared to 48 hpi control), whereas the IFU of PCG- and BTZ-treated *Chlamydia* was kept at a very low number (0.003 ± 0.002% compared to 48 hpi control; Figure 5f, right). We further confirmed that BTZ treatment strictly kills persistently *C. trachomatis* L2-infected HEp-2 or A549 cells. However, the killing efficiency of BTZ treatment against persistently *C. trachomatis* D-infected cells were weaker than that of persistently *C. trachomatis* L2-infected cells (Appendix A). Next, it was verified that other proteasome inhibitors also killed persistently *Chlamydia*-infected host cells. Interestingly, all proteasome inhibitors, except BTZ, even IXA, did not kill host cells harboring persistently infected *Chlamydia* (Figure 5g,h).

### 2.6. BTZ Induces Persistently Chlamydia-Infected Cell Apoptosis through Caspase-3 Activation and Following Pathway

To elucidate how BTZ induces cell death, specifically in persistently *Chlamydia*-infected cells, this study examined the activation of apoptosis-related factors in PCG- and BTZ-treated cells (Figure 6a). At 3 hpi, the cleaved caspase-3 levels in cells were similar to uninfected cells (Figure 6b, lane 2). In contrast, the amount of both caspase-3 activation and its substrate protein poly(ADP-ribose) polymerase (PARP) cleavage significantly increased at 48 hpi, suggesting that cell death had begun for the releasing of infectious EBs (Figure 6b, lane 3). For 48 hpi, PCG-treated infected cells showed little caspase-3 activation and the following PARP cleavage (Figure 6b, lane 4). In contrast, caspase-3 activation and a marked increase in PARP cleavage were observed in PCG- and BTZ-treated infected cells (Figure 6b, lane 5). In addition, at 24 h after the removal of chemicals (recovery 24 h), enhanced activated caspase-3 was observed in PCG- and BTZ-treated infected cells (Figure 6b, lane 7). Furthermore, the amount of cleaved caspase-7 increased only in recovery 24 h in PCG- and BTZ-treated infected cells (Figure 6b), suggesting cell death by specific apoptosis. Next, this study examined whether persistently *Chlamydia*-infected cell apoptosis reduces the amount of *Chlamydia* during the recovery phase. It was confirmed that the addition of pan-caspase inhibitor z-VAD-FMK dramatically increased the IFU scores of PCG- and BTZ-treated infected cells (6.48 ± 2.85%) during the recovery phase (Figure 6c), which was 2160 times higher than that without z-VAD-FMK treatment (compared to Figure 5f). This result strongly indicated that cell-specific apoptosis plays an important role in controlling the chlamydial recovery from a persistent state. 

## 3. Discussion

This study investigated the possibility of anticancer drugs BTZ and IXA being used as antichlamydial agents. BTZ and IXA fulfill the three structures, a membrane-permeable cap, a hydrophobic dipeptide, and a strong reactive group, which were already proposed to function as antichlamydial agents [12]. Surprisingly, BTZ and IXA inhibited chlamydial growth at IC_50_ at the nanomolar level ~400-fold more potently than MDL-27170 and z-FA-FMK (Figure 1), as reported previously [12]. Especially, BTZ showed an IC_50_ of 24 nM against *C. trachomatis*, comparable to the antichlamydial activity of tetracycline (IC_50_ = 22 nM) [20]. BTZ is essentially a proteasome inhibitor, and its terminal boronyl group is covalently bound to the hydroxy group of the 20S proteasome’s threonine residue [19]. This suggested that the peptidyl boronate of BTZ is thought to be responsible for this potent inhibitory activity against *Chlamydia*. Other nondipeptide backbone proteasome inhibitors, such as MG132, did not inhibit chlamydial growth or were rather slightly promoted (Figure 2). This result strongly suggested that BTZ does not suppress chlamydial growth by inhibiting the host cell proteasome; rather, BTZ directly affects the chlamydial factor to inhibit its growth. However, chlamydial growth was immediately recovered after BTZ removal (Figure 3h). In addition, the EM analysis revealed that the BTZ-treated chlamydial RBs apparently had normal morphology (Figure 4). These findings suggested that BTZ bacteriostatically inhibits chlamydial growth by arresting their binary replication of RBs but not by bringing cells to persistent infection. The question that remains to be answered is how BTZ inhibits the binary replication of chlamydial RBs.

Recently, BTZ was identified by screening for inhibitors against the caseinolytic protease (Clp) complex in *Mycobacte**ria*, and BTZ was also reported to inhibit the growth of *Mycobacterium tuberculosis* [21]. Clp is a bacterial proteasome-like complex that degrades unrequired or unfolded proteins [22]. In fact, both the Lon protease and Clp complex are estimated to carry out ~80% of the cellular proteolysis in bacteria [23,24]. Therefore, dysfunction of the Clp complex can be a serious problem for bacterial survival. To back that up, ClpP2 knockdown in *C. trachomatis* causes significant growth retardation [25,26]. This report strongly suggested that BTZ specifically inhibits the chlamydial Clp complex and affects their growth. Unfortunately, this study failed to prove this hypothesis due to the instability of the recombinantly purified ClpP protein (data not shown). The Clp complex is highly conserved in many bacteria, and Clp in *E. coli* and *S. aureus* are strongly suppressed by BTZ in vitro. Nevertheless, BTZ did not inhibit the growth of these bacteria in the experiments, even at 50 μM (Figure 3b-d). It is still unknown for the proper explanations for these findings, but one possibility is that the intracellular bacteria have a more limited source of nutrients than the extracellular bacteria and are much more dependent on the recycling of unwanted proteins (i.e., the Clp complex). These possibilities may be clarified in the future by measuring the exact level of dependence on the Clp complex for the nutrient requirements in *Chlamydia.*

One of the most important findings in this study is that BTZ selectively induces the apoptosis of persistently *C. trachomatis*-infected host cells. After a limited-time treatment with BTZ, only the infected cells with persistent *Chlamydia* inside were dead, whereas uninfected cells still survived (Figure 5d). In this case, the underlying mechanism of these phenomena seems to be completely different from the mechanism of action of BTZ on the proliferative *Chlamydia* RB described above. Our results demonstrated that the killing efficiency of the BTZ treatment on persistently *C. trachomatis* D-infected cells was less than that of L2-infected cells (Appendix A). This is because each treatment condition (such as the BTZ concentration or time course) in this study was optimized for the infection system using HeLa cells and *C. trachomatis* L2. Therefore, we considered that further examinations are required to maximize the BTZ effect on each chlamydial strain or host cell line. IXA showed a marked inhibitory effect on proliferative *Chlamydia* (Figure 1) but little killing effect on persistently *Chlamydia*-infected cells (Figure 5). There are some pharmacokinetic or pharmacodynamic differences between BTZ and IXA. For example, the half-life (t_1/2_) of dissociation of IXA from the proteasome is approximately six times shorter than that of BTZ, and the recovery of proteasome activity with IXA-treated cells is faster than BTZ-treated cells [27,28]. These differences in properties may be responsible for the viability of the infected cells.

This study at least confirmed caspase-3 activation and downstream PARP cleavage in BTZ-treated persistently infected cells, and this activation was not observed with PCG treatment alone (Figure 6). As described in Section 2.1, the original function of BTZ is to induce tumor cell apoptosis by suppressing their proteasome. This suggested that cells harboring persistently infected *Chlamydia* may be highly sensitive to the proteasome inhibitory activity of BTZ. Dean and Powers reported that persistent *C. trachomatis* infection causes resistance to apoptotic stimuli [29]. It is a plausible scenario that BTZ can “unlock” this apoptotic resistance by using an unknown mechanism, which may well be related to proteasome inhibition and lead to cell death. However, the exact mechanism of persistent infected cell-specific killing by BTZ remains to be elucidated, necessitating further studies.

Drug repositioning is a research method to discover new drug effects from existing drugs whose safety and pharmacokinetics in humans have already been confirmed [30]. A number of drugs have already been created through drug repositioning, such as aspirin for antiplatelet agents and thalidomide for multiple myeloma [30]. The greatest advantages of drug repositioning are the certainty of the confirmed safety and pharmacokinetics at the clinical level and the low cost of using a large amount of the existing data. This study demonstrated the potential of BTZ as a treatment for chlamydial infectious diseases through drug repositioning. In particular, a new direction of treatment for persistent *Chlamydia* infection was proposed, which eliminates the infected cells specifically induced by apoptosis. Of course, because BTZ is an anticancer drug, its direct use in treating *Chlamydia* infections is not realistic due to its side effects. Hence, it is necessary to discover drugs that do not affect the human body but are effective only for *Chlamydia* against various derivatives based on BTZ in the future. 

## 4. Materials and Methods

### 4.1. Reagents

The compounds used in this study were purchased as follows. BTZ was from FUJIFILM Wako Pure Chemical. MLN9708 (IXA citrate), delanzomib, and z-FA-FMK were from Cayman Chemical. MDL-28170 was from Merck. Proteasome inhibitors MG132, MG115, and PSI were from the PEPTIDE Institute. Carfilzomib was from Adipogen. z-VAD-FMK was from Bachem. These compounds were separately dissolved in DMSO (Nacalai tesque) at 10 mM and diluted with Dulbecco’s Modified Eagle’s Medium (FUJIFILM Wako Pure Chemical). 

The antibodies used in this study were purchased as follows. Rabbit anti-human cleaved caspase-3 (Asp175) polyclonal antibody (#9661), rabbit anti-human caspase-7 antibody (#9492), rabbit anti-human cleaved PARP (Asp214) polyclonal antibody (#9541), goat anti-rabbit IgG horseradish peroxidase (HRP)-linked antibody (#7074), and horse anti-mouse IgG HRP-linked antibody (#7076) were from Cell Signaling Technology. Mouse anti-β-actin (C4) monoclonal antibody (sc-47778) was from Santa Cruz Biotechnology.

### 4.2. Bacterial Culture, Infection, and Chemical Treatment

*C. trachomatis* serovar L2 (strain 434, ATCC VR-902B) and *C. trachomatis* serovar D (strain: UW-3/Cx, ATCC VR-885) were propagated in HeLa cells, as described previously [31]. For the experiment, the host cells were plated in 24-well tissue culture dishes and infected with *Chlamydia* at the indicated multiplicities of infection (MOI). After inoculation, the plate was centrifuged for 1 h at 900× *g* at room temperature (RT). Each chemical was added to the growth medium at the indicated times and concentrations. For the induction of persistent chlamydial infection, the infected cells were treated with 5 units/mL of PCG (Meiji Seika) at 3 or 6 hpi, and then, PCG was removed to recover the chlamydial growth. Cycloheximide (1 μg/mL) was added in all chlamydial infection experiments in this study.

To obtain bacterial growth curves, *E. coli* BL21, *S. aureus* NCTC 10442, and *K. pneumoniae* subsp. *pneumoniae* were separately grown overnight in LB broth at 37 °C and 180 rpm and then diluted in fresh LB with or without BTZ at the indicated concentrations, the optical density adjusted at 600 nm (OD600) to ~0.1 each and incubated at 37 °C and 180 rpm, and the OD600 was measured every hour for 9 h.

### 4.3. Fluorescent Immunostaining and Microscopy

In most cases, cells on coverslips were fixed with ice-cold methanol for 5 min and stained with a FITC-conjugated mouse monoclonal anti-*Chlamydia* multiepitope antibody cocktail containing Evans blue counterstaining dye (PROGEN) at RT for 30 min. For live/dead cell staining, the cells were incubated with a Zombie NIR^TM^ fixable viability kit (BioLegend) at RT for 30 min before fixation. A fluorescent dye, 4′,6-diamidino-2-phenylindole (DAPI), was used when nuclear staining was needed. Fluorescence images were acquired using a Zeiss Axioskop fluorescence microscope and Zeiss LSM710 confocal microscope with Zeiss ZEN 2010 acquisition software.

### 4.4. Inclusion Forming Unit Assay

For evaluations of the production of chlamydial infectious progeny (Figure 1h, Figure 2b, Figure 5f and Figure 6c), *C. trachomatis*-infected cells were harvested at 48 hpi using 1 mL of sucrose-phosphate-glutamate (SPG) buffer, collected in a 1.5-mL tube and stocked at −80 °C. Cells were disrupted by freeze–thawing following sonication using a VC50PB ultrasonic processor for 10 s. (Sonics & Material, Inc., Newtown, CT, USA). The resulting cell lysates were appropriately diluted with SPG buffer, inoculated on the HeLa cell monolayer, and centrifuged for 1 h at 900× *g*. After being centrifuged, the SPG buffer was changed to the culture medium and cultured for 24 h. For the counting of chlamydial inclusion, the cells were fixed with ice-cold methanol and stained with the FITC-conjugating anti-*Chlamydia* multiepitope antibody. The chlamydial inclusions were counted under a fluorescence microscope, and the IFU score of each sample was calculated.

### 4.5. Cytotoxicity Assay

Cell cytotoxicity was measured using a CytoTOX 96^®^ Non-Radioactive Cytotoxicity Assay kit (Promega), according to the manufacturer’s protocol. As a 100% cytotoxic control, the cells were lysed with 0.1% Triton X-100 in a culture medium and used.

### 4.6. Electron Microscopy

*Chlamydia*-infected HeLa cells were harvested at the indicated postinfection times and fixed with 2% glutaraldehyde overnight, followed by 1% osmium tetroxide (OsO4) at 4 °C for 2 h. The fixed samples were dehydrated by graded ethanol extraction and propylene oxide and embedded with EPON 812 epoxy resin (TAAB Laboratories Equipment) for 96 h. Thin sections (70 nm) were prestained with 5% uranyl acetate and poststained using a lead citrate solution for 5 min before imaging on a Hitachi HT7800 transmission electron microscope.

### 4.7. Western Blotting

Cell lysates were separated by sodium dodecyl sulfate–polyacrylamide gel electrophoresis and transferred to a polyvinylidene difluoride membrane (Bio-Rad, Hercules, CA, USA). Transferred membranes were then blocked for 1 h at RT with 5% (*w*/*v*) skim milk (BD Difco^TM^ 232100). Antibodies against the target proteins were used for antigen detection. HRP-conjugated antibodies were used for secondary antibodies. Antigen–antibody complexes were revealed with the EzWestLumi plus detection reagent (ATTO Corp, Tokyo, Japan) and ImageQuant^TM^ LAS4000 Mini Biomolecular Imager (GE Healthcare, Chicago, IL, USA). The images were analyzed with ImageJ 1.53k software (National Institutes of Health, Bethesda, MD, USA). 

### 4.8. Statistical Analysis

Statistical analysis was performed using Microsoft Excel and GraphPad Prism 6.0. *p* < 0.05 was considered statistically significant.

## Figures and Tables

**Figure 1 ijms-23-07434-f001:**
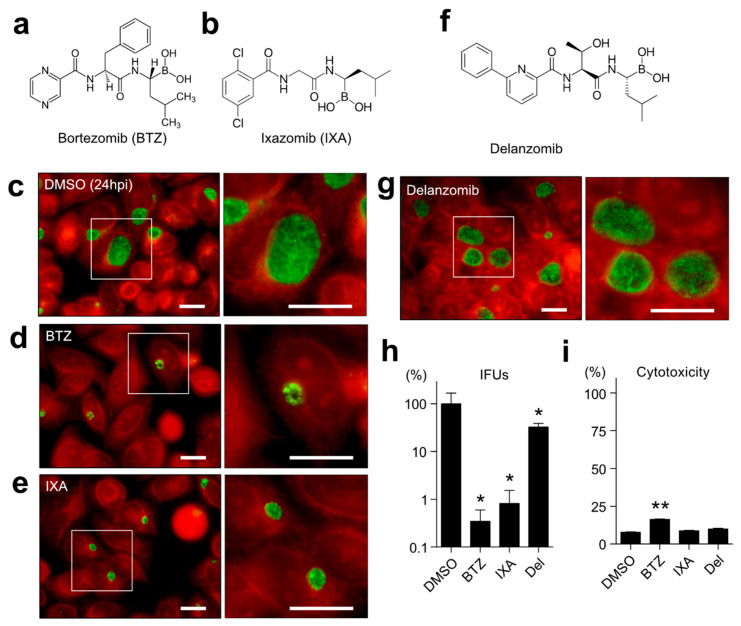
BTZ and IXA, but not delanzomib, showed antichlamydial effects. (**a**,**b**) Chemical structures of BTZ (**a**) and IXA (**b**). (**c**–**e**) *Chlamydia trachomatis* L2-infected HeLa cells were treated with 0.1% DMSO (**c**), 1 μM BTZ (**d**), or 1 μM IXA (**e**) for 24 h. Cells were fixed with ice-cold methanol and stained using a FITC-conjugated anti-*Chlamydia* LPS antibody. The red stain denotes Evans blue counterstaining. Each right panel represents the inset of a high-magnification image of chlamydial inclusion. Scale bar, 20 μm. (**f**) The chemical structure of delanzomib. (**g**) Delanzomib-treated chlamydial inclusion. The inset is a high-magnification image. Scale bar, 20 μm. (**h**) Each sample was harvested at 48 hpi, sonicated, and reinfected onto HeLa cells. The numbers of inclusion were counted, and the inclusion-forming units (IFUs) were calculated. (**i**) The HeLa cells were separately treated with the indicated chemicals for 48 h (1 μM each). Each culture supernatant was collected, and the released LDH activity was measured. The total cell lysate was used as a 100% cytotoxicity control. Data are the mean ± standard deviation (SD) of three independent wells. * *p* < 0.05 and ** *p* < 0.01 compared to each control sample by Welch’s *t*-tests.

**Figure 2 ijms-23-07434-f002:**
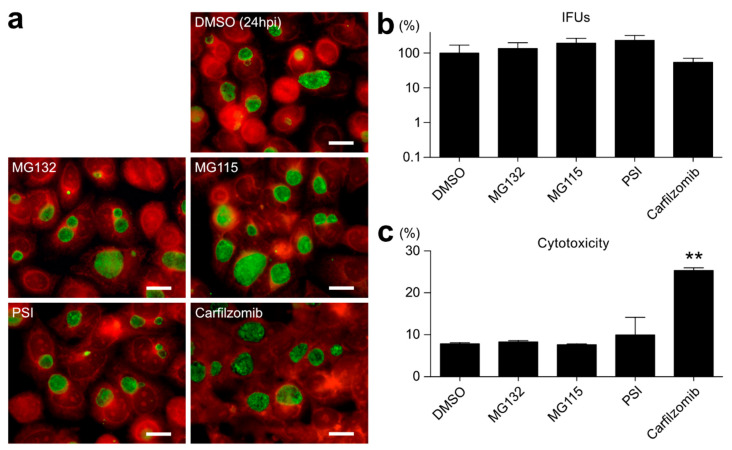
Proteasome inhibitors other than BTZ do not inhibit chlamydial growth. (**a**) *C.*
*trachomatis* L2-infected cells were separately treated with an indicated proteasome inhibitor (1 μM each) for 24 h and fixed and immunostained as described in Figure 1. A red stain denotes Evans blue counterstaining. Scale bar, 20 μm. (**b**) Infected cells were harvested at 48 hpi, and then, the IFU scores were evaluated. (**c**) Cytotoxicities of used chemicals for HeLa cells. Cells were separately treated with chemicals for 48 h (1 μM each), and the released LDH activity was measured as described in Figure 1i. Data are the mean ± *SD* of three independent wells. ** *p* < 0.01 compared to each control sample by Welch’s *t*-tests.

**Figure 3 ijms-23-07434-f003:**
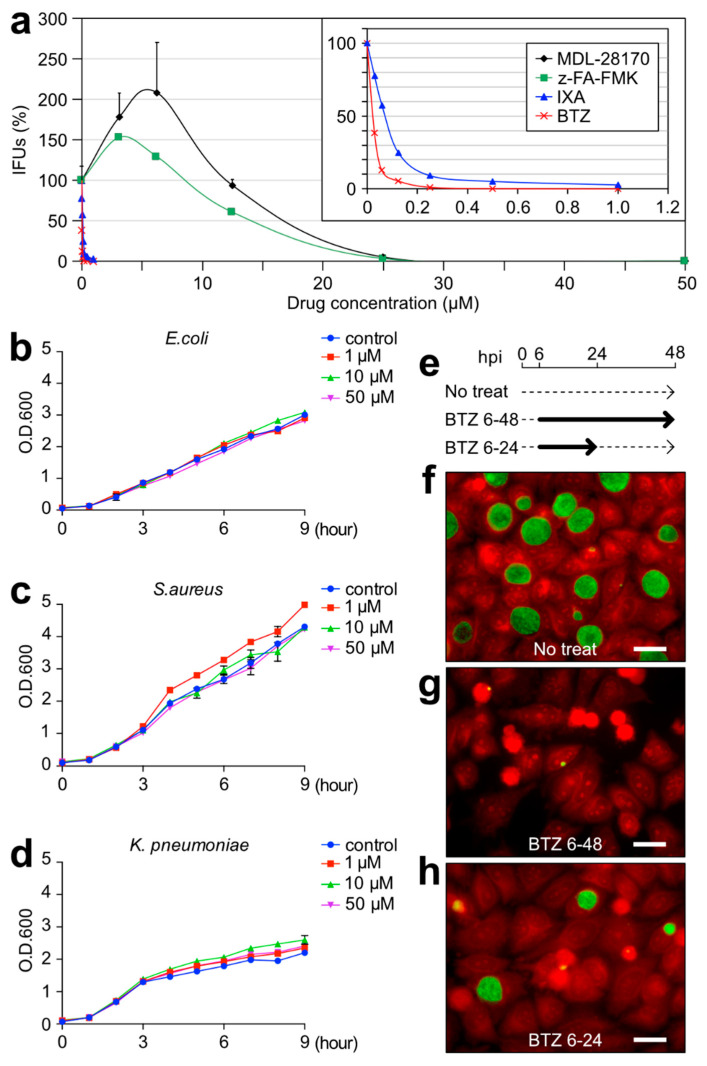
BTZ specifically showed an intense bacteriostatic effect against *Chlamydia*. (**a**) Growth inhibition activity of the indicated four compounds against *C. trachomatis* L2. The infected cells were separately treated with compounds at six- or seven-point concentrations for 48 h, and after, the IFU scores were verified. The inset represents the expansion of the graph from 0 to 1.2 μM. Data are the mean ± SD of three independent wells. (**b**–**d**) Overnight culture of *E. coli* (**b**), *S. aureus* (**c**), and *K. pneumoniae* (**d**) adjusted the turbidity of the optical density at 600 nm (OD600) = 0.1 by diluting it with fresh medium and then culturing it for 9 h with or without BTZ at the indicated concentrations. The turbidity of OD600 nm of each culture was measured. Data are the mean ± SD of three independent cultures. (**e**) Schematic time course of the 1 μM BTZ treatment and removal for *C. trachomatis*-infected cells. Solid arrows denote the treatment period, while dashed arrows denote the culture period without, or removal of, an inhibitor. (**f**–**h**) The infected cells were fixed at 48 hpi and stained as described in Figure 1. A red stain denotes Evans blue counterstaining. Scale bar, 20 μm.

**Figure 4 ijms-23-07434-f004:**
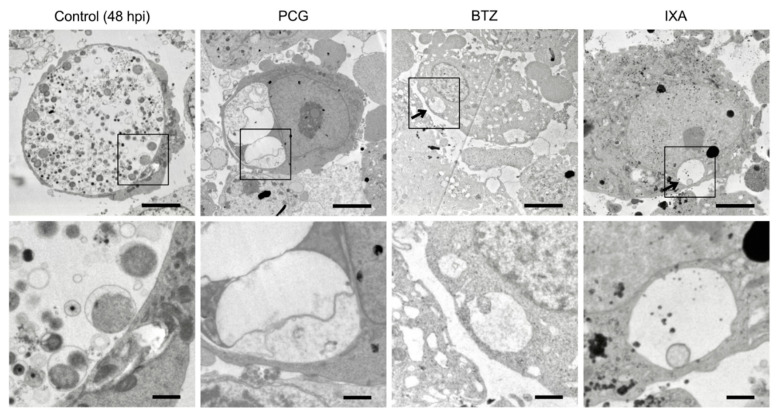
EM analysis of BTZ-treated chlamydial inclusions. (**Top**) *C. trachomatis* L2-infected HeLa cells were separately incubated with BTZ or IXA (1 μM each) or 5 units/mL PCG. All cell samples were harvested at 48 hpi and analyzed by EM (magnification, ×2000). Arrows denote chlamydial inclusion. Scale bar, 5 μm. (**Bottom**) High-magnification image of each black square area. Scale bar, 1 μm.

**Figure 5 ijms-23-07434-f005:**
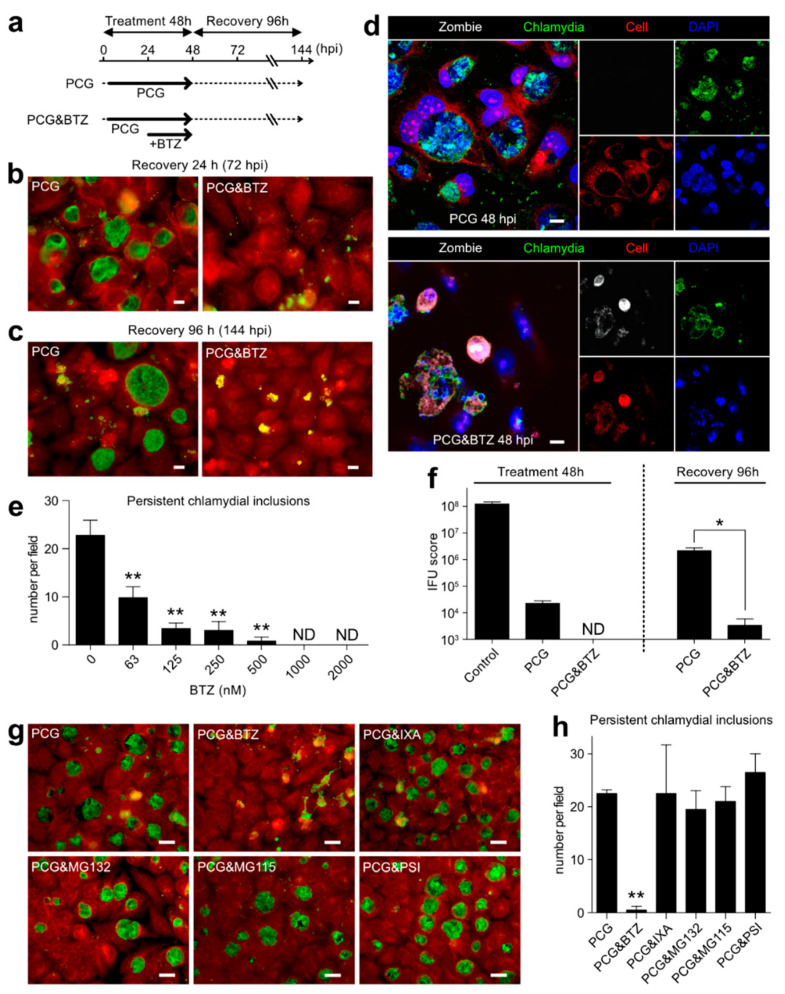
BTZ specifically kills host cells harboring persistently infected *Chlamydia*. (**a**) Schematic diagram of *C. trachomatis* L2 infection and time course of the chemical treatment. (**b**,**c**) The infected cells were treated with PCG or 1 μM BTZ, fixed at 48 hpi (**b**) or 144 hpi (**c**), and stained as described in Figure 1. Scale bar, 20 μm. (**d**) Representative fluorescence images of persistently *Chlamydia*-infected cells living (top) or causing apoptosis by PCG and BTZ treatment (bottom). Scale bar, 10 μm. (**e**) At 72 hpi (recovery 24 h), the cells were fixed, stained, and counted as the live persistently infected chlamydial inclusion per field under a microscope (×40). Data are the mean ± SD of the number of live chlamydial inclusion in five fields. (**f**) The infected cells were treated with indicated chemicals and harvested at 48 hpi (treatment 48 h) or 144 hpi (recovery 96 h), and the IFU scores were evaluated. Data are the mean ± SD of three independent wells. (**g**) The infected cells were treated with PCG for 24 h and then separately added with the indicated chemicals (1 μM each) for a further 24 h. Cells were fixed and stained at 48 hpi. A red stain denotes Evans blue counterstaining. Scale bar, 10 μm. (**h**) The live persistently infected chlamydial inclusions were counted under a microscope. Data are the mean ± SD of number of live chlamydial inclusion in three fields. * *p* < 0.05 and ** *p* < 0.01 compared to each control sample by Welch’s *t*-tests. ND, not detected.

**Figure 6 ijms-23-07434-f006:**
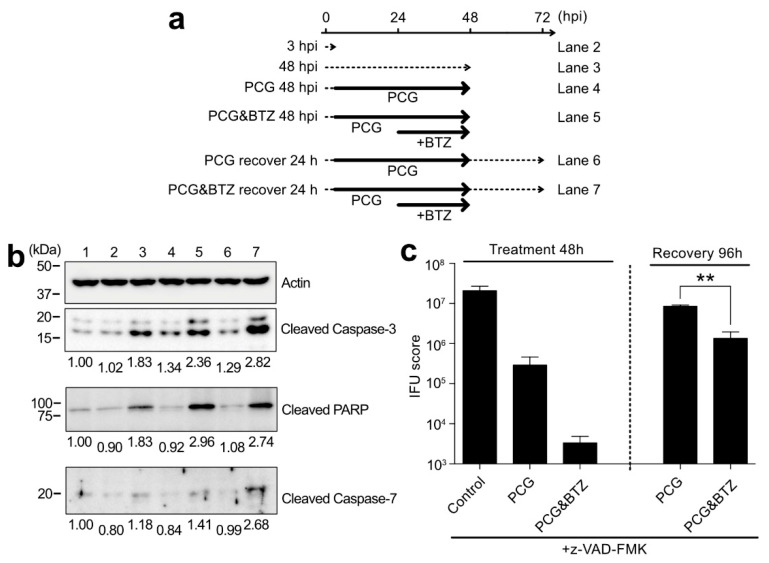
Short BTZ treatment specifically led to persistently *Chlamydia*-infected cell death through caspase activation. (**a**) Schematic of *C. trachomatis* L2 infection, time course of the chemical treatment/removal, and harvest period. (**b**) Each harvested cell sample was analyzed by blotting using a specific antibody. Each band densitometry was normalized with β-actin and defined lane 1 (no infection, time 0) as 1.00. (**c**) The infected cells were treated as described in Figure 5f with 10 μM z-VAD-FMK. The IFU scores were calculated. Data are the mean ± SD of three independent wells. ** *p* < 0.01 compared to each control sample by Welch’s *t*-tests.

## Data Availability

The data presented in this study are available on request from the corresponding author.

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
