# Peer review of "Bortezomib Eliminates Persistent Chlamydia trachomatis Infection through Rapid and Specific Host Cell Apoptosis"

_ijms, 2022, doi:10.3390/ijms23137434_

Round 1

Reviewer 1 Report

The written English/grammar needs reviewing throughout the manuscript. Eg. Line 28 “intracellular pathogens that has the biphasic developmental cycle”; line 47 “completely different from existing antibiotics is always demand”. It is recommended that a native English speaker reviews.

line 38 “not a few cases of treatment failure have been reported. This sentence doesn’t make sense as the authors then go on to describe persistence. Please correct.

Line 45: “world widespread:, please correct.

What is the rationale behind using the L2 serovar instead of one of D-K, that cause common genital infections? L2 infections are not common these days.

Line 83 says there was no significant cytotoxicity, yet the figure shows p<0.01 significant increase in cytotoxicity. Please revise text to accurately reflect data.

Fig 2b and 2c y-axis need labels. What are IFU scores? Please provide details.

More details of methods are required in the methods section. How many fields of view were counted/averaged to calculate fig 1h? This needs to be a substantial number to ensure accurate representation of results. Details of the progeny analysis need including (they are only mentioned in a figure legend).

You need a completely untreated control sample (i.e. media only) for all experiments to confirm/display that the DMSO is also not causing any issues with infection etc.

Fig 3: all organism names on graphs need italicising.

Fig 4: in the BTZ panel the focus of the readers attention is on an area that is not consistent with the other cellular texture, in addition a cell nucleus cannot be seen to confirm that what is being shown is a cell + inclusion. Please provide an image that confirms an inclusion inside a cell (with nucleus present). Please also provide scale bars for the top rows (BTZ also appears to be a different magnification or isn’t cropped as much).

Line 184 and all other places: please correct ‘chlamydial persister’ to persistent chlamydial inclusions.

Line 184: correct ‘decrased’

Line 185: please provide IFU units

Line 203 & 211: correct chlmydia

Line 250: correct Especialy. Please check all spelling throughout the manuscript

Line 335: capital C for Chlamydia

Reviewer 2 Report

This study aims to repurpose two proteasome inhibitors as potential anti-chlamydial drugs. The idea is exciting, the authors propose to inhibit host cell pathways as a new antimicrobial strategy. However, this study has some caveats. The experimental strategy is not optimal, first, some of the experiments are not longitudinal (only one concentration and only one time, figure 1), and the system to quantify bacterial load is based on image analysis. Other options include quantitative PCR for an accurate analysis of this parameter. Bacterial RNA seq will add significative information about the pathways affected in the presence of the proteasome inhibitors. The experiments were performed in HeLa cells, that are physiologically far away from more pertinent environments as primary cells. HeLa cells have 70-90 chromosomes and more than 20 translocations, some of which are highly complex. The text contains numerous errors and wrong phrases. 

Reviewer 3 Report

Dear authors, thank you very much for interesting research article. Though the study seems in their beginning because it rises more questions then give answers, the manuscript seems not a pilot but completed step of study. I'm however curious about the possibility of treatment of dormant forms of Chlamydia Trachomatis or other pathogens with proteasome inhibitor bortezomib (BTZ). The second question of interest is addictive action of BTZ with other antimnicrobial agents, taking into account the example of cumulative effect of the drug with aminoglycosides on Mycobacterium tuberculosis, as noted in article ref#21. Moreover, it is common multidrug therapy concept to use bacteriostatic, plus antibiotic, plus an effective drug with non-specific action (eg., WHO approved anti-leprosy MDT). The aforementioned issues shouldn't be reflected in the manuscript considered, but may investigated in your future research.

Besides a grammar misprint Lon protese at line 268 should be corrected.

Author Response

June 25, 2022

Dear Reviewer 3,

We wish to express our appreciation to you for your insightful comments on our paper. We feel the comments have helped us significantly improve the paper. We have addressed your comments with point-to-point responses and revised the manuscript accordingly.

Response to Reviewer 3 comments

>Dear authors, thank you very much for interesting research article. Though the study seems in their beginning because it rises more questions then give answers, the manuscript seems not a pilot but completed step of study.

Our response: We appreciate the reviewer's favorable evaluating on this point. As you indicated, we are understanding that there are still many questions to be answered in this study. We will continue to resolve these remained issues.

>I'm however curious about the possibility of treatment of dormant forms of Chlamydia Trachomatis or other pathogens with proteasome inhibitor bortezomib (BTZ).

Our response: This study demonstrated that persistently Chlamydia-infected host cells were killed by BTZ treatment. These data do not imply that BTZ directly kills the dormant form of Chlamydia. However, we hope that this new perspective of eliminating the infected host cells may provide a guidepost for treating persistent Chlamydia infection.

>The second question of interest is addictive action of BTZ with other antimicrobial agents, taking into account the example of cumulative effect of the drug with aminoglycosides on Mycobacterium tuberculosis, as noted in article ref#21. Moreover, it is common multidrug therapy concept to use bacteriostatic, plus antibiotic, plus an effective drug with non-specific action (eg., WHO approved anti-leprosy MDT).

Our response: As the reviewer is pointing out, the combination of BTZ with other antibiotics may produce a synergistic effect against chlamydial growth, as in the case of M. tuberculosis. We would like to consider this possibility as soon as possible.

>The aforementioned issues shouldn't be reflected in the manuscript considered, but may investigated in your future research.

Our response: We again appreciate the reviewer for helpful suggestions for future research.

>Besides a grammar misprint Lon protese at line 268 should be corrected.

Our response: Thank you for pointing out. We have corrected this part.

Thank you again for all the thoughtful comments and suggestions that you kindly provided. The authors found them valuable not only for augmenting the scientific preciseness of the manuscript but also for directing our future research.

Sincerely yours,

Ryota Itoh, Ph.D.

Round 2

Reviewer 1 Report

The paper reads much better with the English editing, thank you (I appreciate how hard it is when English is a second lagnuage).

The heading for 2.6 has a spelling mistake (chlmydia)

It is very unfortunate and disappointing that the journal has not given you enough time to complete experiments that involve proper controls. Thank you for providing preliminary data to show there's no difference in IFU's.